# Point Prevalence Survey of Acute Hospital Patients with Difficulty Swallowing Solid Oral Dose Forms

**DOI:** 10.3390/pharmaceutics16050584

**Published:** 2024-04-25

**Authors:** Anne Harnett, Stephen Byrne, Jennifer O’Connor, Eimear Burke, Laura South, Declan Lyons, Laura J. Sahm

**Affiliations:** 1University Hospital Limerick, Dooradoyle, V94 F858 Limerick, Ireland; laura.south@hse.ie (L.S.); declan.lyons@hse.ie (D.L.); 2Pharmaceutical Care Research Group, School of Pharmacy, University College Cork, T12 YN60 Cork, Ireland; stephen.byrne@ucc.ie (S.B.); 119312883@umail.ucc.ie (J.O.); 119348971@umail.ucc.ie (E.B.); 3Pharmacy Department, Mercy University Hospital, Grenville Place, T12 WE28 Cork, Ireland

**Keywords:** solid oral dosage form (SODF), medicine administration, difficulty swallowing, dysphagia, medicine manipulation, inpatient

## Abstract

The safe administration of solid oral dose forms in hospital inpatients with swallowing difficulties is challenging. The aim of this study was to establish the prevalence of difficulties in swallowing solid oral dose forms in acute hospital inpatients. A point prevalence study was completed at three time points. The following data were collected: the prevalence of swallowing difficulties, methods used to modify solid oral dose forms to facilitate administration, the appropriateness of the modification, and patient co-morbidities. The prevalence of acute hospital inpatients with swallowing difficulties was an average of 15.4% with a 95% CI [13.4, 17.6] across the three studies. On average, 9.6% of patients with swallowing difficulties had no enteral feeding tube in situ, with 6.0% of these patients receiving at least one modified medicine. The most common method of solid oral dose form modification was crushing, with an administration error rate of approximately 14.4%. The most common co-morbid condition in these patients was hypertension, with dysphagia appearing on the problem list of two (5.5%) acute hospital inpatients with swallowing difficulties. Inappropriate modifications to solid oral dose forms to facilitate administration can result in patient harm. A proactive approach, such as the use of a screening tool to identify acute hospital inpatients with swallowing difficulties, is required, to mitigate the risk of inappropriate modifications to medicines to overcome swallowing difficulties.

## 1. Introduction

The oral route is the most common route for medicine administration [1]. Although solid oral dosage forms (SODFs), such as tablets and capsules, tend to be the most prevalent and preferred, modifications may be required to ease administration or to allow administration via the oral route. SODF modification can be defined as “any alteration of an oral dosage form that can be performed at the point of administration” [2]. These modifications are undertaken to allow medicine administration to patients with swallowing difficulties (SDs) regarding intact SODFs (e.g., crushing tablets or opening capsules) or to aid fractional dosing (the administration of part of an SODF to allow the administration of a lower dose, e.g., splitting tablets).

Many challenges exist around those with SDs, one of them being protecting the safety of the patient. Whilst dysphagia is a medical term used to describe dysfunction in one or more parts of the swallowing apparatus [3], patients may experience difficulty swallowing SODFs in the absence of a formal diagnosis, which may be described as pill aversion [4]. Presented with this challenge, medication modifications may be attempted, such as crushing tablets or opening capsules. This may not be appropriate legally, pharmaceutically, or therapeutically [5]. A recent review found that to optimise oral medicine modification practices, the needs of individual patients should be routinely and systematically assessed and decision-making should be supported by evidence-based recommendations with multidisciplinary input [6].

Many studies that have examined the prevalence of SDs do so in the context of specific cohorts of patients, such as community-dwelling older adults [7], cardiac surgical intensive care patients [8], those with solid cancers [9], temporomandibular joint disorders [10], and older adult inpatients [11].

Reports on the prevalence of SDs in hospital inpatients often report cohorts where the prevalence is likely to be higher, e.g., older adult care wards [11,12,13], and include inpatients in acute hospital settings and nursing home residents together in the study cohort [14]. A systematic search of the literature did not recover any study that reported the prevalence of swallowing difficulties in general medical and surgical acute hospital inpatients alone.

It is reported that 3% of adult inpatients in the United States of America have a diagnosis of dysphagia [15]. A formal diagnosis of dysphagia is not necessary for a patient to report difficulties in swallowing SODFs and pill aversion [4]. We consider that the true prevalence of acute hospital inpatients with difficulties in swallowing SODFs is likely to be higher than this.

The prevalence of difficulty swallowing SODFs is reported as 29.5% on older adult inpatient units in France [11]. Similarly, the prevalence of inpatients with difficulties in swallowing medicines in a care-of-the-older-adult ward or stroke unit at each of four acute hospitals in the east of England was reported as 34.2% [13]. In a systematic review, 10–34% of inpatients in hospitals, nursing homes, and long-term-stay units had difficulty swallowing SODFs [16]. The prevalence of swallowing difficulties in general medical and surgical acute hospital inpatients alone could not be established from the review of these studies.

The aim of this study was to establish the prevalence of swallowing difficulties with SODFs in hospital inpatients in an acute hospital in Ireland.

## 2. Materials and Methods

### 2.1. Study Design

This is a point prevalence survey (PPS) of swallowing difficulties in acute hospital inpatients.

### 2.2. Ethical Approval and Data Privacy

The study received ethical approval from the Research Ethics Committee, University Hospital Limerick in June 2022 (PPS1) and again in June 2023 (PPS2 and PPS3).

### 2.3. Study Setting

University Hospital Limerick is a model 4 hospital located in the Mid-West of Ireland with a catchment area of 410,000 people [17]. Model 4 refers to a hospital that admits undifferentiated acute medical and surgical patients, including tertiary referred patients, and has a category 3 intensive care unit on site and a 24 h emergency department [18].

### 2.4. Inclusion Criteria

Inpatients with an age greater than 18 years and hospitalised in a ward at University Hospital Limerick by 8 a.m. each day of the survey were eligible for inclusion.

### 2.5. Exclusion Criteria

Outpatients, cancer services patients in the inpatient or day care cancer services wards, critical care patients in the intensive care unit, high-dependency unit, or coronary care units, patients in the psychiatric unit, day patients defined as those discharged on the same day, and inpatients in the paediatric wards were all excluded. One ward with eligible patients (29 beds) was not included in PPS1 because there was a declared outbreak of infection in the ward and access for data collection could not be justified.

### 2.6. Data Collection

The survey data were collected over three time periods. The first PPS was completed in September 2022 (PPS1), the second (PPS2) was completed in June 2023, and final data collection (PPS3) was completed in July 2023.

Data were collected in accordance with a modified version of the methodology established by the World Health Organisation for conducting antimicrobial point prevalence studies [19]. Patients were identified from a ward census.

For all surveys, the following data were collected:Total number of patients identified as eligible for inclusion from ward census;Total number of patients included in the survey.

### 2.7. Procedure

Age and sex were collected for all patients. Patients were interviewed by a research assistant to determine their swallowing status. The following question was asked: “do you have any difficulty swallowing your medicines?” Should a patient confirm that they did have a swallowing difficulty, even in the absence of a formal diagnosis, they were classified as having swallowing difficulties (SDs). In cases where patient interview was not possible, the swallowing status information was obtained from nursing staff. Swallowing status was a binary outcome (yes/no).

For any patient with a swallowing difficulty, the following data were collected: (i) the route of administration of oral medicines and (ii) description of swallowing difficulty for those patients not receiving their medication via an enteral feeding tube and those patients where patient interview was possible or, if relevant, (iii) presence and type of feeding tube. For PPS2 and PPS3, the following additional data were collected: data on their prescribed medicines and disease state(s). The method of SODF modification was obtained from the nursing staff responsible for administering the medicine.

Data were collected in hard copy for each patient and then transferred to Microsoft Excel^®^ 2017 version 2403. All data were stored securely to ensure restricted access and full compliance with General Data Protection Regulations.

Patients with swallowing difficulties were referred to a clinical pharmacist. The pharmacist then provided input into patient care regarding alternative formulations, e.g., suspensions versus tablets.

### 2.8. Data Analysis

Data were analysed using Microsoft Excel^®^ 2017, SPSS version 28 (IBM, Corp. Armonk, NY, USA) and Open Epi [20]. Median and range were reported for age, as data were not normally distributed. Association between categorical variables was assessed. A Pearson’s Chi-square test was conducted to assess whether sex and swallowing difficulties were related. A Mann–Whitney U test was performed to examine whether age differed by a patient’s ability to swallow their SODF medicines. *p*-values < 0.05 were considered to be statistically significant.

An SODF was defined as a product listed by the European Medicines Agency (EMA) as an “oral preparation—solid form” in the EMA list of pharmaceutical dosage forms, with the exception of chewable tablets, which were excluded from the definition in this study.

Each method of SODF manipulation was checked against the Summary of Product Characteristics (SmPC) to assess its appropriateness. If the method used to modify the medicine was not permitted/not described as per the SmPC, then two practice guidelines, (i) Drug administration via Enteral Feeding Tubes [21] and (ii) The NEWT Guidelines for administration of medication to patients with enteral feeding tubes or swallowing difficulties [22], were consulted. Modification practices that did not adhere to the terms of either the SmPC or practice guidelines were considered inappropriate. For these modifications, a medication administration error rate was calculated as follows: number of inappropriate modifications to SODFs/total number of instances in which SODFs were modified, similar to the error calculation method used in other studies [11,12].

If deemed appropriate, the following details were collected:

(i) method of modification, (ii) vehicle used, (iii) number of instances of modification, and (iv) reasons for inappropriateness.

Disease states were classified according to the International Classification of Diseases and Related Health Problems (ICD-10) [23].

## 3. Results

### 3.1. Prevalence of Patients with Swallowing Difficulties

A total of 1120 patients, 96.9% of eligible patients, participated in the PPSs (Table 1).

Of those with swallowing difficulties (n = 172) the median age (years), age range (years), and percentage female were as follows: PPS1 (74, 24–95, 37.8%), PPS 2 (77, 21–93, 49.3%), PPS3 (73, 21–93, 55.2%). The relationship between sex and swallowing difficulties was not significant χ^2^ ([1], *N* = [1120]) = [1.56], *p* = [0.211]. However, a Mann–Whitney U test revealed a statistically significant difference in the age of those with swallowing difficulties (median age 74 with 95% confidence interval CI [73, 77], n = 172) and those without swallowing difficulties (median age 70 with a 95% CI [69, 72], n = 948), U = 69,571, z = −3.064, *p* = 0.002, r = −0.09, although the effect size is small as per the Cohen (1988) criteria [24].

The prevalence of swallowing difficulties in eligible acute hospital inpatients was 13.7% with a 95% CI [10.4, 17.9], 18.0% with a 95% CI [14.5, 22.2], and 14.2% with a 95% CI [11.1, 17.9], in PPS1, PPS2, and PPS3, respectively, with an average prevalence of 15.4% with a 95% CI [13.4, 17.6], (Table 2). On average, 5.8% of patients with swallowing difficulties had an enteral feeding tube in situ (Table 2).

The majority of patients with swallowing difficulties and an enteral feeding tube received their medicines via the tube: PPS1 22/24 (92%), PPS2 17/19 (89.5%), and PPS3 18/22 (82%). Among the patients with a swallowing difficulty and an enteral feeding tube who did not receive their SODFs via the enteral feeding tube (n = 6), three patients swallowed their medicines without modification and three had their medicines crushed and administered in yoghurt. The remaining two patients were not prescribed SODFs and swallowed oral liquid medicines.

The prevalence of patients with swallowing difficulties and an enteral feeding tube requiring modification of their SODFs is recorded in Table 3, with an average prevalence of 5.3%. A patient with a swallowing difficulty and a gastrostomy tube was more likely to have medicines administered via the tube than if the patient had a nasogastric tube (Table 3).

The prevalence of patients with swallowing difficulties and no enteral feeding tube requiring modification of their SODF is recorded in Table 4, with an average prevalence of 6.0%. Regarding patients with swallowing difficulties without an enteral feeding tube, PPS2 and PPS3 show that approximately half of these patients have no modification to their SODF, with 46% and 47.2%, respectively, while in PPS1, all patients with a swallowing difficulty and no enteral feeding tube required some medicines to be altered as a result. PPS2 and PPS3 record that approximately one third of patients with swallowing difficulties and no enteral feeding tube (30% and 30.6%, respectively) require all SODF to be modified, with PPS1 increasing that number to three quarters (77%). In all three PPSs, approximately one fifth of patients with swallowing difficulties and no enteral feeding tube required the modification of some, but not all, of their SODF.

### 3.2. Description of Swallowing Difficulties

Patients were asked to describe their swallowing difficulties if they were not receiving their medicines via an enteral feeding tube and if a patient interview was possible. The majority of patients did not know their medications by name and could not name SODF(s) that they found difficult to swallow. Many gave a vague description like “the large white ones” or “the stomach tablets”, with patients describing the difficulty using tablet sizes and textures rather than the name of the medication (Table 5). Fifty-eight patients provided a description of their swallowing difficulty (PPS1 (n = 5), PPS2 (n = 30), PPS3 (n = 23)). The most common description provided by patients was of difficulty in swallowing “large tablets”, with approximately one third of patients with swallowing difficulties describing difficulty with swallowing all SODFs.

### 3.3. Methods of Solid Oral Dose Form Modification

The most common method used to modify SODFs prior to administration to patients with swallowing difficulties, with or without an enteral feeding tube (Table 6), was crushing.

The most common vehicle used to administer modified SODFs to patients with swallowing difficulties and no enteral feeding tube in all three studies was yoghurt, while water was the most common vehicle to administer modified medicines to patients with swallowing difficulties and an enteral feeding tube.

### 3.4. Appropriateness of Solid Oral Dose Form Modification

The appropriateness of each SODF modification was then established for those patients with swallowing difficulties who had given consent.

In PPS2, twenty-two patients with swallowing difficulties were included, equating to 337 prescriptions (average: 15.3 per patient, range: 8–28). Nine patients received at least one modified SODF.

There were 187 different medications prescribed in PPS2, and 15.5% (n = 29) of these were modified prior to administration. As a medication may be administered more than once daily, we calculated this to be equivalent to 43 instances of medicine modification.

For these 43 instances, 74.4% were not in compliance with the SmPC and 14% were not in compliance with best-practice standards. This gave a medication administration error rate of 14%.

In PPS3, fourteen patients with swallowing difficulties were included, equating to 237 prescriptions (average: 16.9 per patient, range: 15–21). Four patients received at least one modified SODF.

There were 128 different medications prescribed in PPS3, and 15.6% (n = 20) of these were modified prior to administration. As a medication may be administered more than once daily, we calculated this to be equivalent to 27 instances of medication modification.

For these 27 instances, 77.8% were not in compliance with the SmPC and 14.8% were not in compliance with best-practice standards. This gave a medication administration error rate of 14.8%.

Errors included the crushing of enteric coated preparations, increasing the risk of reduced efficacy or increased side effects, such as with aspirin and omeprazole, destruction of the sustained-release formulation, such as with ranolazine, and a modification that was not recommended due to occupational exposure risk, such as with dutasteride. With 40% of the errors, an alternative, more appropriate pharmaceutical formulation was available. In the remaining cases, an alternative therapeutic option could have been considered.

### 3.5. Diseases and Related Health Problems of Patients with Swallowing Difficulties

In 36 patients with swallowing difficulties, there were a total of 151 conditions diagnosed. Hypertension was the most diagnosed condition (n = 17). Two patients had dysphagia listed in their diagnosed health conditions.

## 4. Discussion

These point prevalence surveys established the prevalence of swallowing difficulties in adults with an age greater than 18 years admitted as acute inpatients to general medical and surgical wards in an acute hospital. Across the three surveys, the prevalence of swallowing difficulties among adult hospital inpatients was 15.4% or approximately one in every seven inpatients. Excluding inpatients with an enteral feeding tube in situ, the prevalence of swallowing difficulties was 9.6% or approximately 1 patient in every 10.

A one-day prospective observational study of inpatients (n = 526) with swallowing difficulties in 17 geriatric units (acute geriatric care, rehabilitation unit, long-term care) of the three Paris-Sud teaching hospitals reported an overall prevalence of 29.5%, with a prevalence of 12.2% in the acute care unit [11]. The prevalence in the acute care unit was similar to the prevalence in acute inpatients in this study, with the overall prevalence being much higher than this study when the other settings are included. This is to be expected, as geriatric inpatients in a long-term care or rehabilitation setting are more likely to have a swallowing difficulty [25]. Patients were identified as having a swallowing difficulty through observation by the researchers, which differs from the method used here. It is not clear from the study whether patients with enteral feeding tubes were included, making it difficult to directly compare these populations.

A two-day prospective, observational study of inpatients with an age greater than 65 years (n = 719), in 23 geriatric units in Rouen University Hospital Centre (acute geriatric medicine, post-acute rehabilitation, nursing homes, long-term care units) reported a prevalence of 18.8%, excluding patients with enteral feeding tubes [12]. Our study found a much lower prevalence of swallowing difficulties in hospital inpatients without an enteral feeding tube (9.6%) than this. However, the populations differ in that only those greater than 65 years were included, and some patients were in long-term care, where it would be expected that a higher prevalence of swallowing difficulties would be found [26]. Furthermore, it is unclear from the study how patients with swallowing difficulties were identified for inclusion.

An undisguised direct observational study of inpatients (n = 625) in a care-of-the-older-adult ward or stroke unit at each of four acute hospitals in the east of England over a 4-month period reported a prevalence of 34.2% including those with an enteral feeding tube or 26.2% excluding enteral feeding tube patients [13]. Patients were identified as having a swallowing difficulty if (i) there was advice on fluid, food, or medicine consistency available or (ii) there was an enteral feeding tube in situ, (iii) the nurse considered that the patient had a swallowing difficulty, or (iv) the patient chewed SODFs. The prevalence was much higher than that identified in our study because patients in older adult care wards and stroke wards are more likely to have swallowing difficulties [26], so the two cohorts are not directly comparable.

A comparison of prevalence with other studies is not straightforward because of differences in the populations selected for inclusion, the methods used to identify eligible patients, and whether or not patients with enteral feeding tubes, which can be used for medicine administration, were included in the swallowing-difficulty cohort. An interesting finding in this study is that just over half of the patients with a swallowing difficulty and no enteral feeding tube had their SODF modified by a nurse prior to administration. However, when a patient is discharged home, a nurse is unlikely to be involved in the administration process and the patient may find it difficult to swallow the medicine without modification. Several authors report that patients are at risk of not taking medications that are difficult to swallow [27,28,29]. Hence, patients with swallowing difficulties are potentially at risk of intentional non-adherence when discharged from the acute hospital setting [4]. Additionally, only two of the thirty-six patients with swallowing difficulties had dysphagia recorded in their disease states, potentially making it challenging for health care professionals to be aware of these difficulties. This finding supports the proactive use of a screening tool such as Swallowing Difficulties with Medication Intake and Coping Strategies (SWAMECO) [30]. The SWAMECO questionnaire can be used to identify patients with difficulty swallowing SODFs. Whilst it was originally developed for patients with systemic sclerosis, it was subsequently validated in community-dwelling adult patients [31]. The current version of SWAMECO (version 5) contains 18 questions, 11 of which can be answered with “Yes” or “No”. It includes questions such as “Does your doctor know about your swallowing difficulties when taking medicines?” (question 4) and “Which strategies do you use to make it easier to swallow medicine(s)?” (question 9). This questionnaire can be completed by the patient in approximately five minutes.

In keeping with other studies, the most common method of SODF modification was crushing [11,12,13,32,33,34,35]. Other, less common, methods have also been previously reported, such as capsule opening [35], tablet splitting [13,34,35], and mixing the SODF with food following crushing or opening [36]. This study also found that soaking of the SODF in yoghurt prior to administration, to soften it, was used as a method of administration. Yoghurt was the most popular food stuff used to administer modified SODFs. The use of yoghurt as a vehicle has also been previously reported [11,26,28]. The appropriateness of yoghurt as a vehicle has not been studied, as far as the authors are aware. Over half of the patients with swallowing difficulties reported that their difficulty occurred with large tablets, although the term “large tablet” is not defined and, as the patients could not identify their medicines by name, it is not possible to establish whether all patients with difficulties with large tablets were referring to similar-sized tablets. Size, shape, colour, surface characteristics, taste, and mouthfeel are reported to influence the ease with which an SODF can be swallowed [27,28,37]. In this study, size was the primary descriptor used to report swallowing difficulties with SODFs. The modification of SODFs prior to administration can lead to medication administration errors [13,32]. This study found an error rate of approximately 14.4%. Other studies report error rates from 3.1% [38] to 48.2% [11]; however, direct comparison of medication administration error rates in patients with swallowing difficulties with other studies is difficult due to variation in the cohort included in terms of age, setting, and the presence of an enteral feeding tube. Direct comparison is also complicated by variation in the method used to calculate the error rate, the description of a medication administration error, and the calculation of a medication administration error rate for the entire patient cohort in the study and not just patients with swallowing difficulties [12,32,39]. A prospective observational study of inpatients in three nursing homes in the Netherlands found a medication administration error rate of 3.1% [38]. This study also reported a reduction in medication administration errors to 0.5% following the introduction of a set of warning labels printed on each patient’s unit dose packaging indicating whether a medication could be crushed, as well as education sessions for staff.

### 4.1. Implications for Practice

This study reports that a minimum of one in every seven adult acute hospital inpatients have difficulties in swallowing SODFs, with approximately one third of those having an enteral feeding tube for medicines administration. Whether SODFs are administered via an enteral feeding tube or not, swallowing difficulties lead to SODF modification, most often by crushing tablets or opening capsules [13]. Medication administration errors occur when SODFs are manipulated to facilitate oral administration in patients with swallowing difficulties [39]. This can have catastrophic consequences for individual patients [40]. Studies have shown that the inappropriate manipulation of SODFs, such as crushing sustained release tablets, decreases when guidelines on administration are available and followed [12] and when advice labels are used [38].

### 4.2. Strengths and Limitations

This study was completed at three different time points, with consistent findings in relation to study outcomes enhancing confidence in the results. However, all data were collected from the acute inpatient population of a single hospital. The study results would have been strengthened if multiple hospitals had been included in the PPS.

### 4.3. Recommendations

All acute hospitals should screen patients for difficulties in swallowing SODFs at the point of admission and have administration guidelines available to those who administer medicines to inpatients with difficulties in swallowing. Practice with regard to the manipulation of medicines to facilitate SODF administration to patients with swallowing difficulties should be audited to ensure that practices are safe for the patient. Training should be offered to staff in relation to prescribing and administering SODFs to adult acute hospital inpatients with swallowing difficulties, and, considering the risks associated with medication administration errors, a multidisciplinary approach is warranted [41]. It has been shown that administration errors due to inappropriately crushing tablets can be significantly reduced by using warning symbols as part of the labelling system in conjunction with education [38]. Additionally, it has been reported that significant and sustainable quality improvement in medication administration in nursing home residents with swallowing difficulties can be achieved following the implementation of a programme that includes education, the introduction of a protocol and pocket cards, the screening of medicines by pharmacy technicians, and the annotation of charts with advice on crushing [33]. Additionally, some medicines, including those with antimuscarinic activity and calcium channel blockers, are considered to potentially induce dysphagia, something that also needs to be recognized at patient reviews [26].

## 5. Conclusions

The results of this point prevalence show that at least one in seven adult inpatients in acute hospitals may have difficulties in swallowing medicines. Systems in acute hospitals need to be aware of this prevalence in order to identify these patients, to allow swallowing difficulties to be considered when prescribing, dispensing, and administering SODFs. To allow a proactive pragmatic approach to medicine administration in these patients, we suggest the use of a screening tool to identify them at the point of admission, allowing targeted advice regarding the administration of their medications. This will ensure that the risk of medication administration error is minimised and patient adherence is maximised.

## Figures and Tables

**Table 1 pharmaceutics-16-00584-t001:** Demographics of participating inpatients.

Title of Survey	Number of Patients Eligible for Inclusion	Number of Participating Inpatients (%)	Median Age (Range) Years	Female Sex (%)
PPS1	348	328 (94.3%)	72 (18–99)	45.43%
PPS2	390	383 (98.2%)	70 (18–99)	45.43%
PPS3	418	409 (97.8%)	71 (18–99)	49.14%
Total	1156	1120 (96.9%)	71 (18–99)	

**Table 2 pharmaceutics-16-00584-t002:** The prevalence of adult acute hospital inpatients with swallowing difficulties.

Title of Survey	Number PSDs (%)	Number PSDs and No EFT (%)	Number PSDs and EFT (%)	Type EFT
PPS1	45/328 (13.7%)	21/328 (6.4%)	24/328 (7.3%)	8 G, 16 NG
PPS2	69/383 (18%)	50/383 (13.0%)	19/383 (5.0%)	3 G, 16 NG
PPS3	58/409 (14.2%)	36/409 (8.8%)	22/409 (5.4%)	10 G, 12 NG
Total	172/1120 (15.4%)	107/1120 (9.6%)	65/1120 (5.8%)	21 G, 44 NG

PSDs: patients with swallowing difficulties, EFT: enteral feeding tube, G: gastrostomy tube, NG: nasogastric tube.

**Table 3 pharmaceutics-16-00584-t003:** Prevalence of SODF modification in patients with a swallowing difficulty and enteral feeding tube.

Title of Survey	PSDs and EFT/Total Number PSDs	PSDs and EFT						Prevalence of PSDs and EFT Requiring Modification of SODF
		All SODF Modified		Some SODF Modified		No SODF Modified		
		Gast	NG	Gast	NG	Gast	NG	
PPS1	24/46 (52%)	8/8 (100%)	14/16 (87.5%)	0/8 (0%)	0/16 (0%)	0/8 (0%)	2 ^a^/16 (12.5%)	22/328 (6.7%)
PPS2	19/69 (27.5%)	3/3 (100%)	15/16 (93.8%)	0/3 (0%)	0/16 (0%)	0/3 (0%)	1 ^b^/16 (6.25%)	18/383 (4.7%)
PPS3	22/58 (38%)	9/10 (90%)	10/12 (83%)	0/10 (0%)	1 ^c^/12 (8.3%)	1 ^b^/10 (8.3%)	1 ^b^/12 (8.3%)	20/409 (4.9%)
								60/1120 (5.3%)

^a^ No SODF prescribed, liquid medicines taken orally. ^b^ SODF taken orally. ^c^ Some SODF crushed and taken orally in yogurt, other SODF crushed and administered via the EFT, SODF: solid oral dose form, PSDs: patients with swallowing difficulties, EFT: enteral feeding tube, Gast: gastrostomy tube, NG: nasogastric tube.

**Table 4 pharmaceutics-16-00584-t004:** Prevalence of SODF modifications in patients with swallowing difficulties and no enteral feeding tube.

Title of Survey	PSDs No EFT	PSDs and No EFT			Prevalence of PSDs and No EFT Requiring Modification of SODF
		All SODF Modified	Some SODF Modified	No SODF Modified	
PPS1	21/46 (47.8%)	17/21 (77.3%)	4/21 (18.2%)	0/21 (0%)	21/328 (6.4%)
PPS2	50/69 (72.5%)	15/50 (30.0%)	12/50 (24.0%)	23/50 (46.0%)	27/383 (7.0%)
PPS3	36/58 (62.1%)	11/36 (30.6%)	8/36 (22.2%)	17/36 (47.2%)	19/409 (4.6%)
Average	107/1120 (9.6%)				67/1120 (6.0%)

SODF: solid oral dose form, PSDs: patients with swallowing difficulties, EFT: enteral feeding tube.

**Table 5 pharmaceutics-16-00584-t005:** Patient descriptions of difficulties in swallowing SODFs.

PSD Description of Size or Texture of SODF Contributing to the Swallowing Difficulty (n = 58)	PSDs Having Their Medicines Modified (n = 25)	PSDs Not Having Their Medicines Modified (n = 33)	Total Number of PSDs (Percent)
Capsules	1	1	2 (3.4)
Large tablet(s)/SODF(s)	12	19	31 (53.4)
All SODFs	10	7	17 (29.3)
Small tablet(s)/SODF(s)	1	3	4 (6.9)
“Chalky” SODFs	1	1	2 (3.4)
Not described	0	2	2 (3.4)
Total	25	33	58

SODF: solid oral dose form, PSDs: patients with swallowing difficulties.

**Table 6 pharmaceutics-16-00584-t006:** Methods of SODF modification in PSDs with no EFT.

Method of Modification of SODF	Number of PSDs with No EFT Receiving Modifications to Some or All of Their SODF		
	PPS 1 n = 21 (%)	PPS 2 n = 27 (%)	PPS 3 n = 19 (%)
Crushed	18 (81.8%)	19 (70.3%)	12 (63%)
Capsule opened/pierced	0	1 (3.7%)	0
Split	2 (9%)	5 (18.5%)	5 (26%)
Chewed or halved	0	1 (3.7%)	0
Soaked in yoghurt	1 (4.5%)	1 (3.7%)	2 (10.5%)

SODF: solid oral dose form, PSDs: patients with swallowing difficulties, EFT: enteral feeding tube.

## Data Availability

The ethics application in this study neither sought nor received permission to share these data publicly.

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
