# Peer review of "Point Prevalence Survey of Acute Hospital Patients with Difficulty Swallowing Solid Oral Dose Forms"

_pharmaceutics, 2024, doi:10.3390/pharmaceutics16050584_

Round 1
Reviewer 1 Report
Comments and Suggestions for Authors
Dear authors,
Thanks for the well written article that I had the chance to review. Please find below a few recommendations to further improve your manuscript.
-
The term “elderly (for instance line 55) should no longer be used, see the following reference: Reference: Avers, Dale DPT, PhD; Brown, Marybeth PT, PhD, FAPTA; Chui, Kevin K. PT, DPT, PhD, OCS, GCS; Wong, Rita A. PT, PhD; Lusardi, Michelle PT, DPT, PhD. Use of the Term “Elderly”. Journal of Geriatric Physical Therapy 34(4):p 153-154, October/December 2011. | DOI: 10.1519/JPT.0b013e31823ab7ec. Please replace it for instance with the term “older adults”, which is also used by Health authorities nowadays.
-
What was the time difference between the second and third survey (June 2023 and July 2023)? How have you ensured that you did not have the same patients in both surveys?
-
In the method section you mentioned that the description of the swallowing difficulty (line 106) was noted. However, I don’t see any information on this in the result section. Please comment on this.
-
Have you analyzed the properties of the prescribed medicine (line 108f) in relation to the swallowing difficulties observed, e.g. are swallowing difficulties more common for larger dosage forms? Or for a particular type of SODF?
-
Table 4: → PPS1: I had the feeling that a typo had happened in this line. Shouldn't it be 17/21, 4/21 and 0/21?
-
In line 255f you write that patients with swallowing difficulties that modify their SODF prior to administration are at risk of intentional non-adherence. Can you please explain why?
-
You propose in line 259f that screening tools should be proactively used to find patients with swallowing difficulties and mention the SWAMECO tool. Could you please add a few more sentences to help increase awareness among the readers for such a tool?
-
You use quite a number of abbreviations. While SODF is quite common, others are not (PSD, MAE). Please reconsider the use of abbreviations, or potentially explain it a second time, e.g. MAE in line 270.
Reviewer 2 Report
Comments and Suggestions for Authors
This manuscript is an interesting study of a growing problem as the population ages.
Major changes
1. The authors should compute confidence intervals for the prevalence. This can be easily done with openepi free software. https://www.openepi.com/Proportion/Proportion.htm
2. For the patients' age, as the data do not have a normal distribution, the authors should compute the confidence interval of the median. That can also be done with open epi. https://www.openepi.com/Median/CIMedian.htm
Mino issues
1. In the abstract, 95% confidence intervals should be used for the prevalence.
2. Is this correct .“For any patient (…) if relevant (iv) presence and type of feeding
3. What does it mean the acronym MAE. It has to be explained the first time that is used.
4. Lines 68 there are two dots eliminate one. “inpatients. .”
5. “Age and gender “ Line 98. (Gender means sexual orientation implies also social factors, sex is a biological charasteristic) Did you really register gender (Male, female, non binary…etc)
6. The reader should be able, to understand a table ,without reading the paper. The tables should be designed in a way that can be used in a power presentation or included for example in a book . Due to that all the abreviations and acronyms presents in a table or in the title of the table should be avoided or explained at the foot of the table.
Reviewer 3 Report
Comments and Suggestions for Authors
The Author's study indicates that at least one in seven adult inpatients in acute hospitals may experience difficulty swallowing their medications. As a result, hospital systems must identify these patients and consider their swallowing difficulty when prescribing, dispensing, and administering medications. To address this issue proactively, the study's authors suggest using a screening tool to identify patients with swallowing difficulties at admission, which would allow for targeted advice regarding medication administration.
Minor:
Are there similar studies in other continental countries, i.e., whether the geographical region can affect the output.
The authors could attach a detailed protocol for screening patients with a questionnaire and whether, in addition to the questionnaire, a check by the doctor treating the patient is also required, i.e., propose a survey protocol with possible outcomes and criteria when it is necessary to change the way of drug administration.
Reviewer 4 Report
Comments and Suggestions for Authors
This study is a point prevalence survey aimed at recording patients with difficulty swallowing in a hospital. The manuscript presents the results of this analysis in tabular format.
I have two major concerns about the manuscript:
A. What does this study contribute to the scientific community, and what is its clinical impact? The prevalence of patients with swallowing difficulty is already known in the general population. I attempted to identify novel findings, but was unable to do so.
B. There is not even a simple statistical analysis to attempt to identify any significant differences among patient groups. In a study like this, it would be beneficial to explore relationships between patient characteristics.
Additional important concerns:
1. Why was the study conducted in three different time periods? Was any interim analysis performed? It is unclear why a three-period study was planned.
2. There was no sample size estimation, and no indication of power analysis.
3. The manuscript does not mention stratification based on important aspects of the study population such as age group, disease type, pathological status, etc.
Thank you.
Round 2
Reviewer 2 Report
Comments and Suggestions for Authors
The authors have answered satisfactorily to all my questions.
Reviewer 4 Report
Comments and Suggestions for Authors
Thank you for responding to my comments.
Even though some issues raised cannot be addressed in this manuscript, I feel that this work can be published.